# Repressing PTBP1 fails to convert reactive astrocytes to dopaminergic neurons in a 6-hydroxydopamine mouse model of Parkinson's disease

Weizhao Chen[1,2†], Qiongping Zheng[1,2†], Qiaoying Huang[1,2], Shanshan Ma[1,2*], Mingtao Li[1,2*]

[1]Guangdong Provincial Key Laboratory of Brain Function and Disease, Zhongshan School of Medicine, Sun Yat-sen University, Guangzhou, China; [2]Department of Pharmacology, Zhongshan School of Medicine, Sun Yat-sen University, Guangzhou, China

*For correspondence:
mashsh3@mail.sysu.edu.cn (SM);
limt@mail.sysu.edu.cn (ML)

†These authors contributed equally to this work

**Abstract** Lineage reprogramming of resident glial cells to dopaminergic neurons (DAns) is an attractive prospect of the cell-replacement therapy for Parkinson's disease (PD). However, it is unclear whether repressing polypyrimidine tract binding protein 1 (PTBP1) could efficiently convert astrocyte to DAns in the substantia nigra and striatum. Although reporter-positive DAns were observed in both groups after delivering the adeno-associated virus (AAV) expressing a reporter with shRNA or CRISPR-CasRx to repress astroglial PTBP1, the possibility of AAV leaking into endogenous DAns could not be excluded without using a reliable lineage-tracing method. By adopting stringent lineage-tracing strategy, two other studies show that either knockdown or genetic deletion of quiescent astroglial PTBP1 fails to obtain induced DAns under physiological condition. However, the role of reactive astrocytes might be underestimated because upon brain injury, reactive astrocyte can acquire certain stem cell hallmarks that may facilitate the lineage conversion process. Therefore, whether reactive astrocytes could be genuinely converted to DAns after PTBP1 repression in a PD model needs further validation. In this study, we used *Aldh1l1-CreERT2*-mediated specific astrocyte-lineage-tracing method to investigate whether reactive astrocytes could be converted to DAns in a 6-hydroxydopamine (6-OHDA) mouse model of PD. However, we found that no astrocyte-originated DAn was generated after effective and persistent knockdown of astroglial PTBP1 either in the substantia nigra or in striatum, while AAV 'leakage' to nearby neurons was easily observed. Our results confirm that repressing PTBP1 does not convert astrocytes to DAns, regardless of physiological or PD-related pathological conditions.

## Editor's evaluation

In aggregate, we believe that this study provides important new light on the role of PTB1 repression as a potential therapeutic repair strategy. This paper elegantly utilizes a broad array of approaches to demonstrate, in a toxin-based mouse model of Parkinson's disease, that reactive astrocytes fail to convert into neurons such as dopaminergic neurons in brain areas specifically affected in this common neurodegenerative disorder (i.e. the striatum and the substantia nigra) upon repression of the RNA-binding protein PTB1. This finding is of enormous importance since previous studies have reported the reverse, namely that astrocytes can be converted into dopaminergic neurons in response to the repression of PTB1 and have used these results to propose that inhibition of PTB1 by gene therapy could represent new repair strategies for patients with disorders of the nervous system, such as Parkinson's disease.

## Introduction

The emergence and rapid development of in vivo cell reprogramming technology, which converts deleterious astrocytes into functional neurons, holds great promise for neuroregenerative therapy (*Torper and Götz, 2017*). Various groups worldwide have successfully achieved astrocyte-to-neuron (AtoN) conversion by forced expression of different proneural transcription factors (TFs) such as Neurogenin2 (Ngn2) (*Grande et al., 2013*), Mash1 (Ascl1) (*Liu et al., 2015*), NeuroD1 (*Guo et al., 2014*), SOX2 (*Niu et al., 2013*), and various TF combinations (*Rivetti di Val Cervo et al., 2017*; *Wu et al., 2020*; *Lentini et al., 2021*). In contrast to the TF overexpression approach, by repressing an RNA-binding protein polypyrimidine tract binding protein 1 (PTBP1), two groups have reported that functional neurons including dopaminergic neurons (DAns) could be induced from astrocytes rapidly and efficiently in vivo, reconstructing the nigrostriatal circuit and improving motor deficits in a 6-hydroxydopamine (6-OHDA) mouse model of Parkinson's disease (PD) (*Qian et al., 2020*; *Zhou et al., 2020*).

Nevertheless, without substantiating the exact origin of the nascent-induced DAns using a reliable lineage-tracing strategy, these two outstanding works soon arouse widespread debate and argument (*Jiang et al., 2021*; *Arenas, 2020*; *Qian et al., 2021*). Most recently, by adopting the stringent lineage-tracing method, two studies arguing against previous findings have been published. One group shows that adeno-associated virus (AAV)-sh*Ptbp1*-induced, presumed astrocyte-converted DAns are not truly converted from astrocytes but are merely AAV-infected endogenous neurons due to virus leakage (*Wang et al., 2021*). Another group reports that no astrocyte-derived neuron, including DAns are generated in multiple brain regions including the substantia nigra and striatum, in astrocyte-specific *Ptbp1* deletion mice (*Blackshaw et al., 2021*). However, both studies only focus on quiescent astrocytes and whether reactive astrocytes could be converted to neurons more effectively after PTBP1 repression requires further verification.

During brain injury or neurodegeneration, astrocytes become activated and acquire certain characteristics of neural stem cells (NSCs) such as proliferation, Nestin- or Vimentin-immunoreactivity, and multipotency (*Buffo et al., 2008*; *Shimada et al., 2012*; *Robel et al., 2011*; *Sirko et al., 2013*). Some researchers have claimed that reactive astrocytes with stem cell hallmarks can be reprogrammed to neurons more easily and efficiently than quiescent astrocytes (*Grande et al., 2013*; *Guo et al., 2014*; *Brulet et al., 2017*; *Wan et al., 2014*; *Mattugini et al., 2019*). Therefore, we adopted the 6-OHDA PD model with lineage-tracing method to investigate whether reactive astrocytes could truly be converted to neurons including DAns.

## Results

### Repressing PTBP1 efficiently induces viral-reporter-labeled neurons

To effectively repress astroglial PTBP1 in vivo, we designed and synthesized AAV (serotype 2/5) expressing EGFP, followed by shRNA targeting mouse *Ptbp1* as previously reported (*Qian et al., 2020*), under the full-length (2.2 kb) *GFAP* promoter (AAV-sh*Ptbp1*). The corresponding virus expressing scramble shRNA (AAV-shscramble) was used as a control (*Figure 1A*).

First, to verify the cell specificity of AAV-sh*Ptbp1* and AAV-shscramble, we performed immunostaining and GFP[+] cell counting 7 days after AAV delivery into the substantia nigra and striatum, respectively. Our results showed that both AAVs mostly infected astrocytes (AldoC[+]), but not neurons (neuronal nuclei, NeuN[+]), NG2-glia (NG2[+]), or microglia (Iba-1[+]) (*Figure 1—figure supplement 1*).

Next, to investigate whether repressing astroglial PTBP1 could gradually convert astrocytes to DAns in the substantia nigra and striatum, brain slices of different timepoints (1, 2, and 3 months) after AAV injection were collected for immunostaining analysis. PTBP1 expression was not affected by AAV-shscramble, but was downregulated to undetectable levels by AAV-sh*Ptbp1* in GFP[+] cells from 1 to 3 months (*Figure 1B, C*), indicating astroglial PTBP1 was consistently repressed.

The pan-neuronal marker NeuN and the DAn marker tyrosine hydroxylase (TH) were then co-stained with GFP. The results showed that very few GFP[+]NeuN[+] cells (approximately 2.39%) were detected even at 3 months after AAV-shscramble injection, while remarkable GFP[+]NeuN[+] cells were detected, with 12.65 ± 1.56%, 41.10 ± 1.98%, and 46.09 ± 12.9% GFP[+] cells expressing NeuN, and with 8.99 ± 1.99%, 30.40 ± 3.03%, and 37.93 ± 9.92% expressed TH, at 1, 2, and 3 months after AAV-sh*Ptbp1* injection in the substantial nigra (*Figure 1C, D*). In the striatum, only GFP[+]NeuN[+] cells, but not GFP[+]TH[+]

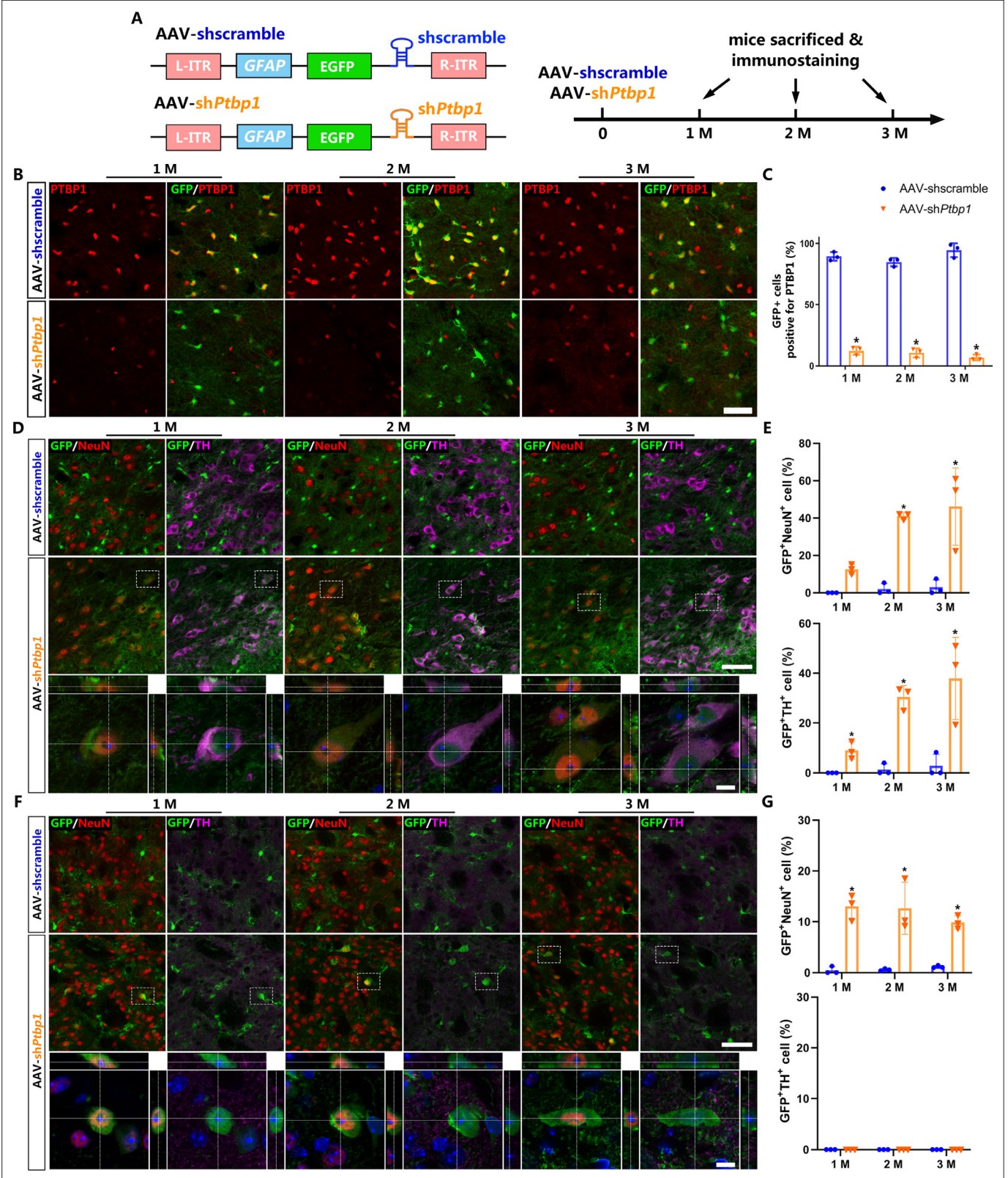

**Figure 1.** Viral-reporter-labeled neurons including dopaminergic neurons are induced in the substantia nigra and striatum after adeno-associated virus (AAV)-sh*Ptbp1* delivery. (**A**) Schematic of AAV-sh*Ptbp1* and AAV-shscramble vector design and the experimental design. (**B**) Representative images of brain slices co-stained with polypyrimidine tract binding protein 1 (PTBP1) (red) and GFP (green) at indicated timepoints after AAV-sh*Ptbp1* or AAV-shscramble delivery in the substantia nigra. Scale bar, 50 μm. (**C**) Quantitative data of GFP+PTBP1+ cells ratio from (**B**) are shown. Representative images

*Figure 1 continued on next page*

*Figure 1 continued*

of brain slices co-stained GFP (green) with tyrosine hydroxylase (TH; purple) or NeuN (red) at indicated timepoints after AAV-sh*Ptbp1* or AAV-shscramble delivery in the substantia nigra (**D**) and striatum (**F**), the enlarged 3D reconstruction of boxed individual neurons are shown in the lower panel (counterstained with Hoechst–blue). Scale bars: low magnification, 75 μm; high magnification, 10 μm. Quantitative data of GFP$^+$NeuN$^+$ or GFP$^+$NeuN$^+$ cells ratio from the substantia nigra (**D**) and striatum (**F**) are shown in (**E**) and (**G**). n = 3 biological repeats per group. Data are presented as mean ± SEM. * indicates a significant difference between AAV-sh*Ptbp1* and AAV-shscramble (p<0.05). Two-way ANOVA followed by Tukey's multiple comparisons test is used. (**C**) $F_{(2,12)}$ = 5.297, 1 M: p<0.0001; 2 M: p<0.0001; 3 M: p<0.0001. (**E**) $F_{(2,12)}$ = 5.321, 1 M: p=0.5220; 2 M: p=0.0016; 3 M: p=0.0007. (**G**) $F_{(2,12)}$ = 1.132, 1 M: p=0.0004; 2 M: p=0.0006; 3 M: p=0.0085.

The online version of this article includes the following source data and figure supplement(s) for figure 1:

**Source data 1.** Brain slices co-stained with PTBP1 (red) and GFP (green) at indicated timepoints after AAV-sh*Ptbp1* or AAV-shscramble delivery in the substantia nigra for *Figure 1B, C*.

**Source data 2.** Brain slices co-stained GFP (green) with TH (purple) or NeuN (red) at indicated timepoints after AAV-sh*Ptbp1* or AAV-shscramble delivery in the substantia nigra for *Figure 1D, E*.

**Source data 3.** Brain slices co-stained GFP (green) with TH (purple) or NeuN (red) at indicated timepoints after AAV-sh*Ptbp1* or AAV-shscramble delivery in the striatum for *Figure 1F, G*.

**Figure supplement 1.** Adeno-associated virus (AAV)-sh*Ptbp1* and AAV-shscramble predominantly infected astrocyte at early timepoint (7dpi).

**Figure supplement 1—source data 1.** Brain slices co-stained with GFP (green) and AldoC (purple), NeuN (red), NG2 (purple), or Iba-1 (red) 7days after AAV-sh*Ptbp1* or AAV-shscramble delivery in the substantia nigra or striatum for *Figure 1—figure supplement 1*.

cells, were detected after AAV-sh*Ptbp1* delivery (*Figure 1E, F*). These results highly resemble results of Qian's study (*Qian et al., 2020*) but against that of Zhou's study (*Zhou et al., 2020*).

However, these results are not sufficient to prove that astrocytes were truly converted to neurons including DAn, as AAV-mediated gene expression could be 'leaked' into neurons as indicated by a recent study (*Wang et al., 2021*). Thus, more solid evidence such as lineage-tracing is needed to verify the exact origin of these viral-reporter-labeled neurons and DAns.

## PTBP1 repression fails to convert quiescent astrocytes to DAns

Genetic lineage-tracing (*Kretzschmar and Watt, 2012*) has been widely recognized as the most convincing strategy for cell source identification and is generally performed by combining cell-specific Cre recombinase-expressing mice with Cre-activated reporter mice. *Aldh1l1-CreERT2* mice with the highest specificity to target astrocytes (*Srinivasan et al., 2016*) were chosen to cross-breed with a reporter mouse *Rpl22^{lsl-HA}* (Ribotag) (*Sanz et al., 2009*), in which the endogenous ribosomal protein Rpl22 was tagged with three copies of the hemagglutinin (HA) epitope after Cre-mediated recombination (*Figure 2A*). After Tamoxifen (TAM)-mediated induction of CreERT2 activity, almost all of the AldoC-positive astrocytes were specifically labeled with the HA epitope (*Figure 2B*) and barely no HA leaky expression was observed in the neurons of the substantia nigra and striatum of *Aldh1l1-CreERT2;Rpl22^{lsl-HA}* mice.

Two weeks after TAM induction, *Aldh1l1-CreERT2:Rpl22^{lsl-HA}* mice were injected with AAV-sh*Ptbp1* and AAV-shscramble into the substantia nigra or striatum to verify whether GFP$^+$TH$^+$ or GFP$^+$NeuN$^+$ cells were originated from HA-labeled astrocytes (*Figure 2C*). Three months later, the mice were sacrificed for triple immunostaining for GFP, HA and NeuN, or for GFP, HA, and TH. Through exhaustive examination of the whole midbrain and striatum, no GFP$^+$TH$^+$ or GFP$^+$NeuN$^+$ cells that were simultaneously HA-positive could be detected in either AAV-sh*Ptbp1* (*Figure 2D, E*) or AAV-shscramble (*Figure 2—figure supplement 1A–B*) treated mice.

Therefore, the lineage-tracing results clearly illustrat that PTBP1 repression fails to convert quiescent astrocytes to neurons including DAns, which is consistent with recent studies (*Wang et al., 2021*; *Blackshaw et al., 2021*).

## PTBP1 repression also fails to convert reactive astrocytes to DAns in a 6-OHDA model

Many studies have suggested that reactive astrocytes may acquire certain characteristics of NSC upon brain injury, which may promote the AtoN conversion process (*Grande et al., 2013*; *Guo et al., 2014*; *Brulet et al., 2017*; *Wan et al., 2014*; *Mattugini et al., 2019*). To verify whether repression of PTBP1 could convert reactive astrocytes to neurons including DAns, we injected AAV-shscramble or

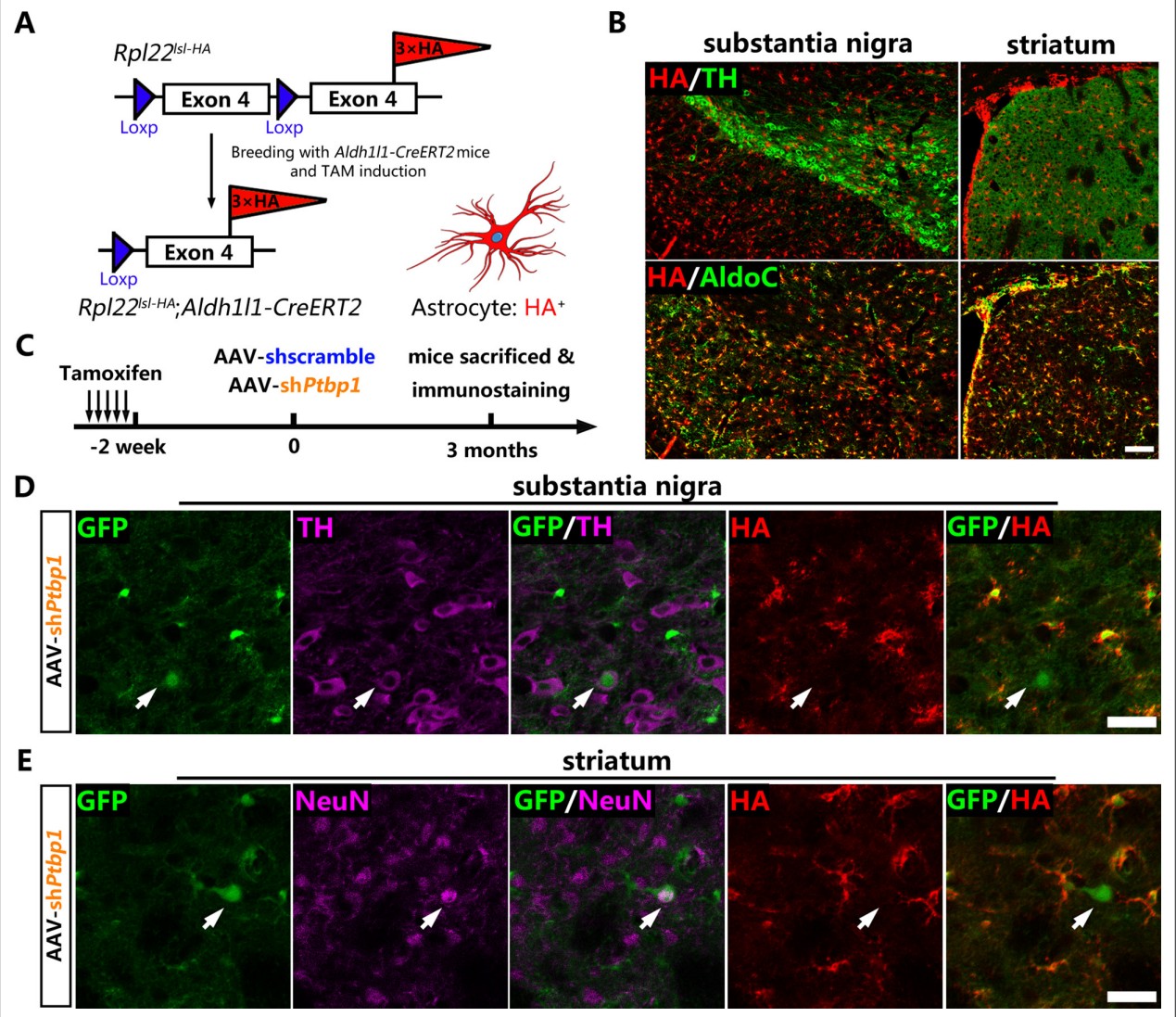

**Figure 2.** No viral-reporter-labeled neuron including dopaminergic neuron is derived from quiescent astrocytes AAV-sh*Ptbp1* delivery. (**A**) Schematic of breeding strategy of *Aldh1l1-CreERT2;Rpl22^{lsl-HA}* lineage-tracing mice. (**B**) Experimental design of Tamoxifen (TAM) induction and representative images of the substantia nigra or striatum of *Aldh1l1-CreERT2;Rpl22^{lsl-HA}* mice co-stained hemagglutinin (HA) (red) with pan-astrocyte marker AldoC (green) and tyrosine hydroxylase (TH) (purple) 2 weeks after TAM administration. Scale bar, 100 µm. (**C**) Schematic of experimental design. Representative images of brain slices co-stained GFP (green), HA (red) with TH (purple) in the substantia nigra (**D**) or with NeuN (purple) in striatum (**E**) 3 months after AAV-sh*Ptbp1* delivery. n = 3 biological repeats per group. Arrows indicate GFP/TH (**D**) or GFP/NeuN (**E**) double positive neurons that are HA negative. Scale bar, 75 µm.

The online version of this article includes the following source data and figure supplement(s) for figure 2:

**Source data 1.** Indicated brain regions of *Aldh1l1-CreERT2;Rpl22^{lsl-HA}* mice co-stained hemagglutinin (red) with pan-astrocyte marker AldoC (green) and tyrosine hydroxylase (purple) 2 weeks after Tamoxifen administration for *Figure 2B*.

**Source data 2.** Brain slices co-stained GFP (green), HA (red) with TH (purple) in the substantia nigra 3 months after AAV-sh*Ptbp1* delivery for *Figure 2D*.

**Source data 3.** Brain slices co-stained GFP (green), hemagglutinin (red) with NeuN (purple) in the striatum 3 months after adeno-associated virus-sh*Ptbp1* delivery for *Figure 2E*.

**Figure supplement 1.** Representative images of brain slices co-stained GFP (green), hemagglutinin (HA; red) with tyrosine hydroxylase (TH; purple) in the substantia nigra (**A**) or with NeuN (purple) in striatum (**B**) 3 months after adeno-associated virus (AAV)-shscramble delivery.

AAV-sh*Ptbp1* in the substantia nigra or striatum of the *Aldh1l1-CreERT2;Rpl22^{lsl-HA}* mice 3 weeks after the 6-OHDA lesion (*Figure 3A*).

Our results showed that 6-OHDA induced severe lesions in the nigrostriatal pathway, characterized by significantly reduced numbers of DAns in the substantia nigra and remarkably decreased densities

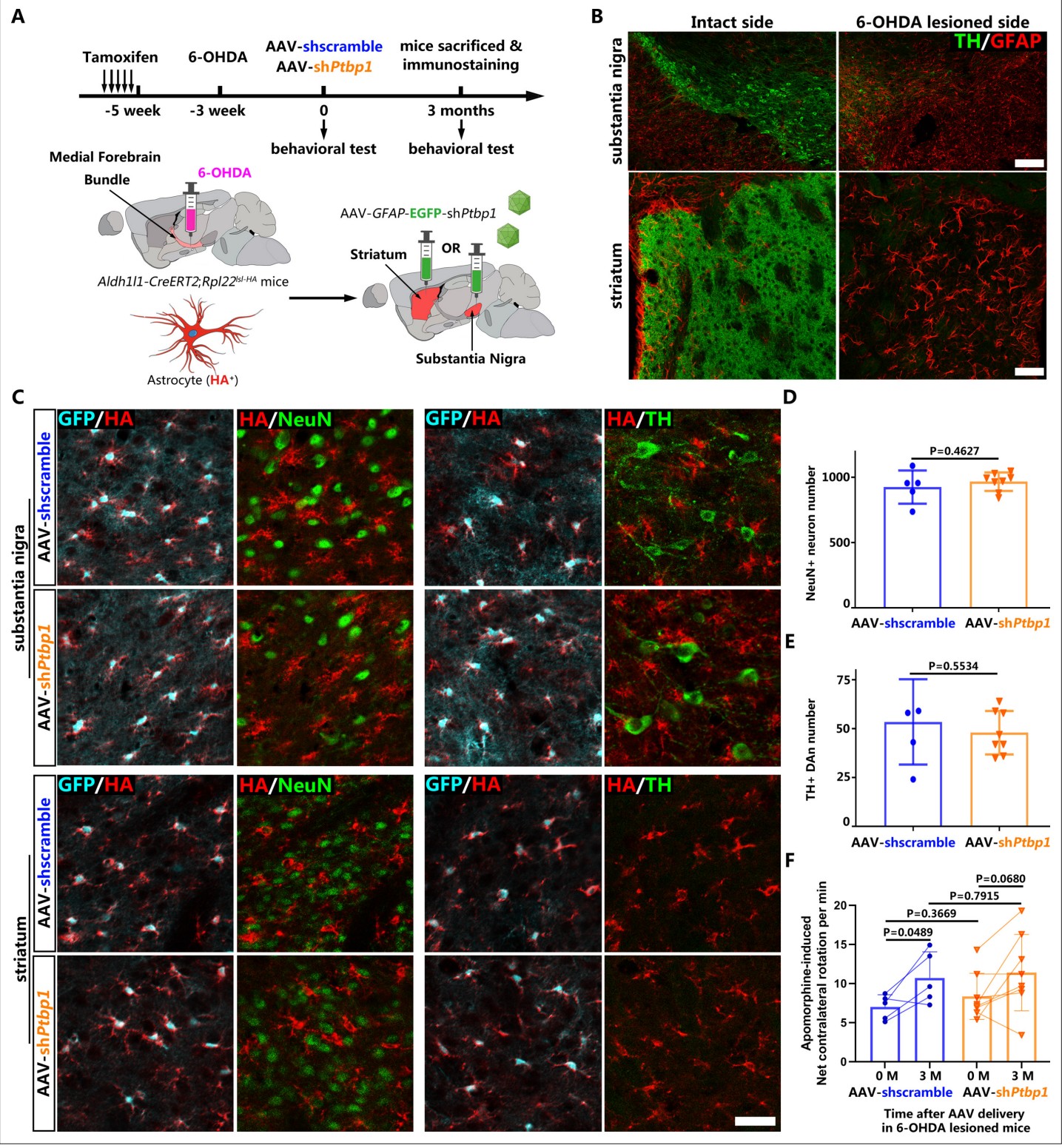

**Figure 3.** No neuron including dopaminergic neuron (DAn) is derived from reactive astrocytes in 6-hydroxydopamine (6-OHDA) model after polypyrimidine tract binding protein 1 (PTBP1) repression. (**A**) Schematic of experimental design. (**B**) Representative images of the substantia nigra or striatum after 6-OHDA lesion co-stained with tyrosine hydroxylase (TH; green) and GFAP (red). Scale bar, 200 μm for the substantia nigra and 50 μm for the striatum. (**C**) Representative images of brain slices of *Aldh1l1-CreERT2;Rpl22^lsl-HA* mice subjected to 6-OHDA lesion and adeno-associated virus (AAV)-sh*Ptbp1* or AAV-shscramble injection in substantia (upper panel) and striatum (lower panel), co-stained with GFP (cyan), hemagglutinin (HA; red) and NeuN (green) or TH(green). Scale bar, 50 μm. Number of NeuN⁺ neurons (**D**) and TH⁺ DAns (**E**) in the substantia nigra 3 months after

*Figure 3 continued on next page*

*Figure 3 continued*

treatment with AAV-sh*Ptbp1* or AAV-shscramble on 6-OHDA lesioned mice. n = 5 mice for AAV-shscramble group; n = 8 mice for AAV-sh*Ptbp1* group. (**F**) Apomorphine-induced rotation test before and 3 months after AAV-sh*Ptbp1* or AAV-shscramble delivery on 6-OHDA lesioned mice. n = 8 mice for AAV-sh*Ptbp1* group; n = 5 mice for AAV-shscramble group. Data are presented as mean ± SEM. Unpaired t test is used in (**D**) $F_{(4, 7)} = 3.266$, p=0.4627 and (**E**) $F_{(4, 7)} = 3.856$ p=0.5534. Unpaired t test is used in (**F**) 0 M AAV-sh*Ptbp1* vs AAV-shscramble $F_{(7, 4)} = 3.59$, p=0.3669; 3 M AAV-sh*Ptbp1* vs AAV-shscramble $F_{(7, 4)} = 2.157$, p=0.7915. Paired t test is used in (**F**), AAV-shscramble 3 M vs 0 M: p=0.0489, df = 4; AAV-sh*Ptbp1* 3 M vs 0 M: p=0.068, df = 7.

The online version of this article includes the following source data for figure 3:

**Source data 1.** Brain slices of the substantia nigra or striatum after 6-OHDA lesion, co-stained with TH (green) and GFAP (red) for ***Figure 3B***.

**Source data 2.** Brain slices of *Aldh1l1-CreERT2;Rpl22^lsl-HA* mice subjected to 6-OHDA lesion and AAV-shPtbp1 or AAV-shscramble injection in substantia or striatum, co-stained with GFP (green) and HA (red), or NeuN (green) and HA (red), or TH(green) and HA (red) for ***Figure 3C***.

**Source data 3.** Original data and statistical analysis of ***Figure 3D and E*** for ***Figure 3D&E***.

**Source data 4.** Original data and statistical analysis of ***Figure 3F***.

of TH$^+$ fibers in the striatum (***Figure 3B***). Meanwhile, astrocytes became remarkably activated, as indicated by classic cytoskeletal and morphological changes, including hypertrophy of the main processes and cell bodies and upregulation of intermediate filament protein GFAP (***Figure 3B***). However, under such circumstance, no NeuN$^+$ neurons including TH$^+$ DAns positive for HA, could be detected, and no obvious morphological changes of astrocytes (indicated by HA staining) were observed after PTBP1 repression (***Figure 3C***), suggesting that neither AtoN nor astrocyte-to-DAn (AtoDAn) conversion occurred. Moreover, the number of NeuN$^+$ neurons (***Figure 3D***) and TH$^+$ DAns (***Figure 3E***) did not increase after AAV-sh*Ptbp1* delivery into the substantia nigra compared with AAV-shscramble delivery. The motor deficits induced by 6-OHDA lesion (reflected by apomorphine-induced rotation) were not improved by AAV-sh*Ptbp1* injection, either (***Figure 3F***).

Together, these data demonstrate that repressing PTBP1 also fails to generate DAns from reactive astrocytes in a mouse 6-OHDA model of PD.

## ASO-mediated PTBP1 repression still fails to convert reactive astrocytes to DAns in a 6-OHDA mouse model of PD

To rule out the possibility that AAV toxicity (*Johnston et al., 2021*) restrained the AtoN conversion process, we synthesized antisense oligonucleotide (ASO) against mouse *Ptbp1* as an alternative strategy for PTBP1 repression (***Figure 4A***). Immunofluorescence results showed that ASO was distributed broadly in the midbrain, as indicated by ASO-attached Cy3, and astroglial PTBP1 was significantly downregulated for 2 months after ASO-*Ptbp1* delivery compared to ASO-Ctrl delivery (***Figure 4—figure supplement 1A, B***). Western blot analysis of the midbrain further confirmed the knockdown efficiency of ASO-*Ptbp1* (***Figure 4—figure supplement 2C, D***). Using brain slices from the same mice, we did not find any astrocyte-originated neurons (YFP$^+$NeuN$^+$) or DAns (YFP$^+$TH$^+$) (***Figure 4B***), suggesting that no neurons, including DAns, were converted from quiescent astrocytes after ASO-*Ptbp1* delivery. Next, we injected ASO-*Ptbp1* or ASO-Ctrl into the substantia nigra of 6-OHDA lesioned *Aldh1l1-CreERT2;Rpl22^lsl-HA* mice (***Figure 4C***). Two months after ASO delivery, we still could not find any neurons including DAns positive for HA (***Figure 4D***), indicating that no neurons, including DAn, were generated from reactive astrocytes. Furthermore, the motor deficits of the 6-OHDA lesioned mice were not alleviated by 2 months of ASO-*Ptbp1* treatment (***Figure 4E***), similar to those of ASO-Ctrl (***Figure 4—figure supplement 2***).

These data demonstrate that repressing astroglial PTBP1 via ASO also fails to generate DAns from either quiescent or reactive astrocytes in a 6-OHDA mouse model of PD.

## Discussion

In this study, through stringent and convincing lineage-tracing technology, we substantiated that neither AAV-shRNA- nor ASO-mediated astroglial PTBP1 repression could achieve AtoN or AtoDAn conversion either in the substantia nigra or in the striatum of a 6-OHDA mouse model of PD.

We first used AAV expressing EGFP followed by sh*Ptbp1* under the full-length *GFAP* promoter to repress astroglial PTBP1 and AAV-shscramble as a control. Both AAV constructs were predominantly

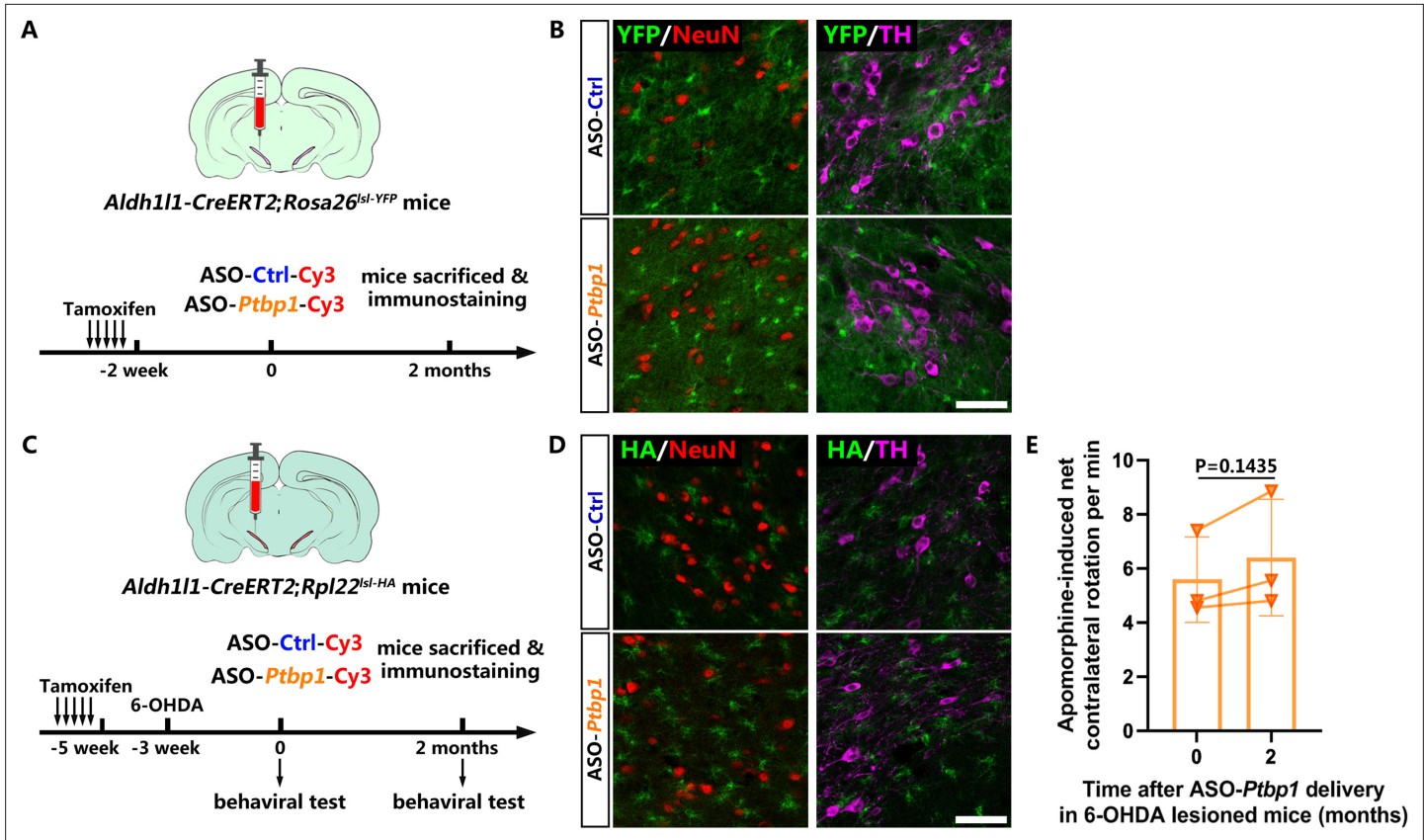

**Figure 4.** No neuron including dopaminergic neuron is derived from astrocytes with or without 6-hydroxydopamine (6-OHDA) lesion after antisense oligonucleotide (ASO)-mediated polypyrimidine tract binding protein 1 (PTBP1) repression. (**A**) Schematic of experimental design. (**B**) Representative images of the brain slices of *Aldh1l1-CreERT2;Rosa26^{lsl-YFP}* mice co-stained with YFP (green) and NeuN (red) or tyrosine hydroxylase (TH; purple) after ASO-*Ptbp1* or ASO-Ctrl delivery in the substantia nigra. Scale bar, 75 µm. n = 6 mice for ASO-Ctrl group; n = 5 mice for ASO-*Ptbp1* group. (**C**) Schematic of experimental design. (**D**) Representative images of brain slices of 6-OHDA lesioned *Aldh1l1-CreERT2;Rpl22^{lsl-HA}* mice after ASO-*Ptbp1* delivery in substantia nigra, co-stained with hemagglutinin (HA; green) and NeuN (red) or TH (purple). Scale bar, 75 µm. n = 2 mice for ASO-Ctrl group; n = 3 mice for ASO-*Ptbp1* group. (**E**) Apomorphine-induced rotation test before and 2 months after ASO-*Ptbp1* delivery in substantia nigra on 6-OHDA lesioned mice (n = 3 biological repeats). Data are presented as mean ± SEM. Paired t test is used in (**E**), p=0.1435, df = 2.

The online version of this article includes the following source data and figure supplement(s) for figure 4:

**Source data 1.** Brain slices of *Aldh1l1-CreERT2;Rosa26^{lsl-YFP}* mice co-stained with YFP (green) and NeuN (red) or TH (purple) after ASO-*Ptbp1* or ASO-Ctrl delivery in the substantia nigra for *Figure 4B*.

**Source data 2.** Brain slices of 6-OHDA lesioned *Aldh1l1-CreERT2;Rpl22^{lsl-HA}* mice after ASO-*Ptbp1* delivery in substantia nigra, co-stained with HA (green) and NeuN (red) or TH (purple) for *Figure 4D*.

**Source data 3.** Original data and statistical analysis of *Figure 4G*.

**Figure supplement 1.** Knockdown efficiency of PTBP1 by ASO.

**Figure supplement 1—source data 1.** Brain slices of *Aldh1l1-CreERT2;Rosa26^{lsl-YFP}* mice co-stained with PTBP1 (purple), YFP (green), and Cy3 (red) after ASO-*Ptbp1* or ASO-Ctrl delivery in the substantia nigra for *Figure 4—figure supplement 1*.

**Figure supplement 2.** Apomorphine-induced rotation test before and 2 months after antisense oligonucleotide (ASO)-Ctrl delivery in substantia nigra on 6-hydroxydopamine lesioned mice (n = 1 biological repeat).

**Figure supplement 2—source data 1.** Original data and statistical analysis of *Figure 4—figure supplement 2*.

expressed in astrocytes 7 days post-infection. After 1–3 months of infection, AAV-sh*Ptbp1* allowed for low levels of reporter protein (GFP) expression in neurons, whereas AAV-shscramble-mediated GFP expression was still restricted to astrocytes. Without stringent lineage-tracing, the result can be easily misinterpreted as PTBP1 repression-mediated AtoN conversion. The reason why AAV-sh*Ptbp1* rather than AAV-shscramble leaked into neurons is currently unclear. According to a recent study, coding sequences of some proneural genes, such as *Neurod1*, could activate *GFAP* promoter elements in cis

and drive the expression of the viral reporter gene in neurons (*Wang et al., 2021*). We presume that the sh*Ptbp1* sequence might function similarly to *Neurod1*, activating *GFAP* promoter and driving GFP expression in neurons at a relatively low level.

It has been reported that AAV can exert toxic effects on doublecortin (DCX)[+] neuroblasts or neural progenitor cells (*Johnston et al., 2021*), which might render the virus-infected astrocytes unable to convert into neurons. After reviewing previous studies of in vivo glia-to-neuron conversion, we found that DCX[+] intermediate cells could hardly be detected using AAV for gene manipulation (*Liu et al., 2015*; *Guo et al., 2014*; *Brulet et al., 2017*; *Mattugini et al., 2019*; *Chen et al., 2020*; *Lai et al., 2020*; *Leib et al., 2022*; *Liu et al., 2020*; *Xiang et al., 2021*; *Zheng et al., 2022*; *Torper et al., 2015*; *Pereira et al., 2017*). In contrast, DCX[+] intermediate cells can usually be seen in most lentivirus (LV)/ retrovirus (RV)-mediated glia-to-neuron conversion studies in vivo (*Grande et al., 2013*; *Guo et al., 2014*; *Niu et al., 2013*; *Buffo et al., 2005*; *Gascón et al., 2016*; *Torper et al., 2013*; *Rivetti di Val Cervo et al., 2017*; *Magnusson et al., 2014*; *Niu et al., 2015*; *Su et al., 2014*; *Wang et al., 2016*; *Zhang et al., 2022*; *Heinrich et al., 2014*). This observation suggests that either AAV toxicity restrains the AtoN conversion process, or LV/RV boosts the AtoN conversion process in an unknown manner.

To exclude the potential toxicity of AAV, we adopted ASO as an alternative method for PTBP1 repression and found ASO-*Ptbp1* also failed to convert astrocytes into neurons or DAns. This result is inconsistent with that of a previous study claiming that AtoN conversion occurred after ASO-*Ptbp1* delivery (*Qian et al., 2020*). However, we believe that their result is not convincing since the reporter mice-*Rosa-Tdtomato* (Ai14) used for lineage-tracing by *Qian et al., 2020* has been questioned for occasional leakage to neurons (*Wang et al., 2021*). Other possible reasons for the discrepancy may be different ASO-mediated PTBP1 repression efficiency, different lineage-tracing mouse types and genetic backgrounds, and different experimental time lengths.

The major weakness of the present study is that we only rule out the possibility of astrocyte conversion to neuron, including DAn using *Aldh1l1* promoter-based lineage-tracing mice. Whether other latent neurogenic cell types such as NSC (*Maimon et al., 2021*), oligodendrocytes (*Weinberg et al., 2017*), or NG2 glia, could be converted to neurons upon PTBP1 repression requires further investigation. Our results showed that ASO had no cell selectivity and could non-specifically enter different cell types to repress PTBP1 expression. These neurogenic cell types could be converted into neurons after PTBP1 repression Therefore, a reliable lineage-tracing method targeting these neurogenic cell types is necessary for future studies to identify the genuine cell identity that might contribute to neuron restoration.

One important question of the present study is whether and to what extent the reactive state of astrocytes in the 6-OHDA model could truly reflect the real state of astrocytes in PD patients. Acute lesions induced by neurotoxins, such as 6-OHDA, usually results in substantial neuron loss, creating an inflammatory microenvironment characterized by the presence of both A1 (pro-inflammatory) and A2 (anti-inflammatory) astrocyte subtypes (*Ryu et al., 2020*). In contrast, as one of the most important risk factors for PD, normal aging induces pro-inflammatory A1 like astrocyte reactivity (*Clarke et al., 2018*), which may accurately reflect the real state of PD patients. In particular, during ischemia stroke, a classic model characterized by the presence of A2 astrocytes, astrocytes spontaneously become neurogenic and the Notch pathway is repressed (*Magnusson et al., 2014*). We therefore assume that pro-inflammatory cytokines, such as tumor necrosis factor-α (TNF-α), could be detrimental to the AtoN conversion process and the survival, maturation, and subsequent neurite outgrowth of the newborn neurons (*Karimi-Abdolrezaee and Billakanti, 2012*; *Belenguer et al., 2021*; *Park and Bowers, 2010*), whereas the anti-inflammatory cytokines and neurotrophic factors, such as brain-derived neurotrophic factor might be beneficial and even critical (*Niu et al., 2013*). Therefore, whether A2 astrocytes could be converted to neurons including DAns more efficiently, and more importantly, how to induce beneficial A2 astrocytes in the brains of PD patients for neural repair and regeneration needs further investigation.

## Materials and methods
### Animals
All animal experiments were performed in accordance with the guidelines of the Institutional Animal Care and Use Committee of University (Approval number: 2018–059). The protocol was reviewed and

approved by the Ethics Committee on Laboratory Animal Care. The mice were housed in rooms with controlled 12 hr light/dark cycles, temperature, and humidity, and food and water were provided ad libitum. Eight- to 10-week-old C57BL/6 mice weighing 22–26 g were obtained from the Vital River Laboratory Animal Technological Company (Beijing, China). *Aldh1l1-CreERT2* transgenic mice (Stock number #029655), *Rpl22 ˡˢˡ⁻ᴴᴬ* (Ribotag) mice (Stock number #011029), and *Rosa26ˡˢˡ⁻ʸᶠᴾ* mice (Stock number #006148) were obtained from The Jackson Laboratory (Bar Harbor, ME, USA). *Aldh1l1-CreERT2* mice were used for breeding to the *Rpl22ˡˢˡ⁻ᴴᴬ* mice or *Rosa26ˡˢˡ⁻ʸᶠᴾ* mice. Eight- to 10-week-old *Aldh1l1-CreERT2;Rpl22ˡˢˡ⁻ᴴᴬ* and *Aldh1l1-CreERT2;Rosa26ˡˢˡ⁻ʸᶠᴾ* mice were used for lineage-tracing experiments.

## Tamoxifen (TAM) administration

The protocol of TAM administration was determined according to previous work (*Srinivasan et al., 2016*) with little modifications. Briefly, TAM-free base (Sigma, Shanghai, China) was dissolved in corn oil (Aladdin, Shanghai, China) at a concentration of 10 mg/mL in a 60°C water bath for 30 mins. TAM was orally administered at a daily dose of 100 mg/kg body weight for 5 consecutive days. Experiments were performed 2 weeks after the last TAM administration.

## 6-OHDA model

The procedure was based on previous study with minor modifications (*Rivetti di Val Cervo et al., 2017*; *Qian et al., 2020*; *Zhou et al., 2020*). In brief, 6-OHDA (Sigma, Shanghai, China) was dissolved in ice-cold saline solution (0.9% NaCl) containing 0.2 mg/mL L-Ascorbic acid (BBI Life Sciences, Shanghai, China) at a concentration of 3 mg/mL. After anesthetized with 3% isoflurane, mice were then placed in a stereotaxic instrument (Model 940, Kopf Instruments, Tujunga, CA, USA) and delivered with 1 µL of 6-OHDA solution (3 µg) into the right medial forebrain bundle (mFB) at a speed of 100 nL/min according to the following coordinates: anteroposterior (A/P) = –1.20 mm, mediolateral (M/L) = 1.30 mm, dorsoventral (D/V) = –4.75 mm. Injections were conducted with a 10 µL syringe connected to a 33-Ga needle (Hamilton, Reno, NV, USA) using a microsyringe pump (KDS LegatoTM 130, Holliston, MA, USA). After 6-OHDA injection, mice were typically allowed to recover for 3 weeks with intense daily care.

## Apomorphine-induced rotation

Apomorphine-induced rotation was performed 3 weeks after 6-OHDA lesion and 3 months after AAVs delivery or 2 months after ASOs delivery. Briefly, 10 mins after intraperitoneal injection of apomorphine (Sigma-Aldrich, 5 mg/kg dissolved in ice-cold saline solution), each mouse was placed in an opaque cylinder (30 cm diameter) for free moving with a camera recording above for 20 mins as reported (*Zhou et al., 2020*).

## AAV production and infection

To effectively repress astroglial PTBP1 in vivo, we designed and synthesized AAV (serotype 2/5) expressing EGFP, followed by shRNA targeting mouse *Ptbp1* (5′-GGGTGAAGATCCTGTTCAATA-3′) as previously reported (*Qian et al., 2020*), under the full-length (2.2 kb) *GFAP* (human glial fibrillary acidic protein) promoter (AAV2/5-*GFAP*-EGFP-5'miR-30a-shRNA(*Ptbp1*)–3'miR-30a-WPREs, AAV-sh*Ptbp1* for short, titer: 3.41E+12 vg/mL). The corresponding virus expressing scramble shRNA (same nucleotide composition but in a different sequence order) was used as a control (AAV2/5-*GFAP*-EGFP-5'miR-30a-shRNA(scramble)–3'miR-30a-WPREs, AAV-shscramble for short, titer: 2.57E+12 vg/mL) (*Figure 1A*). Both AAVs were synthesized based on the pAAV-*GFAP*-EGFP-WPRE-hGH plasmid (Addgene #105549) and packaged by BrainVTA Co., Ltd (Wuhan, China).

Before injection into the mouse brain, the AAVs were adjusted to 1E+12 vg/mL using sterile Dulbecco's phosphate buffered saline (DPBS, Gibco, Thermo Fisher Scientific, Inc, Waltham, MA, USA). Wildtype or lineage-tracing mice were subjected to AAV injection into the substantia nigra (1 µL) or striatum (2 µL), respectively, at a speed of 100 nL/min. The coordinates indicating distance from bregma were A/*P* = –2.90 mm, M/L = 1.30 mm, and D/V = –4.35 mm for the substantia nigra, and A/*P* = 0.80 mm, M/L = 1.60 mm, and D/V = –2.80 mm for the striatum. After injection, the needle remained in place for at least 5 mins to prevent retrograde flow along the needle track, and the needle

was slowly removed from the mouse brain. Cleaning and suturing of the wound were performed after the needle was removed.

## Antisense oligonucleotides (ASOs) synthesis and delivery

ASOs were synthesized by Synbio Technology (Suzhou, China), the sequence and modification for ASO-*Ptbp1* (5′-GTGGAAATATTGCTAGGCAC-3′) and control ASO (5′-CCTATAGGACTATCCAGGAA-3′) were performed as reported (*Maimon et al., 2021*). Briefly, 10 core 2′-deoxyribonucleotides in the central were flanked on both 5′ and 3′ sides by 5 2′-methoxyethyl (MOE)-modified nucleotides. The backbones of all ASOs contain phosphorothioate modifications and all cytosine residues were modified as 5′-methylcytosines. Cyanine dye Cy3 was attached to the 3′ end of those ASOs for fluorescence detection. After dissolved in sterile and Rnase-free DPBS at a concentration of 1 µg/µL, ASOs were subpacked and stored at –80°C to avoid repeated freezing and thawing. A 2 µL of ASO was injected into the substantia nigra (A/P = –2.90 mm, M/L = 1.30 mm, and D/V = –4.35 mm) of astrocyte-specific lineage-tracing mice with or without 6-OHDA lesion.

## Immunofluorescent analysis

For immunofluorescent analysis, mice were anesthetized with 1.25% Avertin and were perfused intracardially with ice-cold phosphate buffered saline (PBS), followed by 4% paraformaldehyde (PFA, Sigma, China) in PBS at a flow rate of 10 mL/min. The brains were then removed and post-fixed in 4% PFA at 4°C overnight (8–12 hrs), followed by immersion in 20 and 30% sucrose for 24 hrs respectively. Immunofluorescent analysis was performed as previously described (*Yu et al., 2018*; *Hu et al., 2019*). In brief, cryostat-coronal sections encompassing the entire midbrain (20 µm) and striatum (30 µm) were serially collected. Free-floating sections were pre-incubated in blocking solution containing 5% normal donkey serum and 0.3% Triton X-100 in 50 mM Tris-buffered saline (pH = 7.4) at room temperature for 1 hr. Primary antibodies against HA tag (Rabbit, Abcam, ab9110, 1:1000), GFP (Rabbit, Abcam, ab290, 1:1000), GFP (Chicken, Abcam, ab13970, 1:1000), TH (Chicken, Millipore, AB9702, 1:1000), NeuN (Mouse IgG1, Millipore, MAB377, 1:1000), and PTBP1 (Rabbit, Invitrogen, PA5-81297, 1:1000) were dissolved in diluent and incubated with sections overnight at 4°C. After washing three times, sections were incubated with the secondary antibodies (Thermo Fisher or Jackson ImmunoResearch), which were conjugated with Alexa 488, Alexa 555, or Alexa 647 at room temperature for 1 hr. The sections were visualized under a confocal laser scanning microscope (LSM 780, Carl Zeiss, Germany) and captured in gray scale and pseudocolored for presentation. A 12 µm thick confocal Z-stack was acquired using ×63 objective. Three-dimensional reconstruction of z-stack images was generated using the Zeiss Zen software (blue edition).

## Immunoblot analysis

Two months after ASO-Ctrl or ASO-*Ptbp1* delivery, mouse midbrains were homogenized, lysed, and resolved using 10% SDS-PAGE, then transferred to a polyvinylidene difluoride (PVDF) membrane and probed with the primary antibodies (PTBP1, Rabbit, Invitrogen PA5-81297, 1:1000; β-actin, Mouse, Santa Cruz sc-47778, 1:5000) overnight at 4°C on a shaker. The secondary antibodies were horseradish peroxidase (HRP)-conjugated, and the signals were detected using ECL.

## Statistics

GraphPad Prism (GraphPad software, version 9.0) was used for the statistical analysis. All data are presented as mean ± SEM (standard error of the mean). When comparing data from two groups, a two-tailed Student's t test was used. When there were two variables, ANOVA followed by Tukey's multiple comparisons test was used. For all analyses, statistical significance was considered when probability value of $p < 0.05$.

## Acknowledgements

We thank Qingxing Zhang for his assistance with the animal breeding and genotyping. This study was supported by grants from the National Key R&D Program of China (2018YFA0108300), the National Natural Science Foundation of China (U1801681, 81771368, 31871019, 32070959), the Key Realm

R&D Program of Guangdong Province (2018B030337001), the Guangdong Provincial Key Laboratory of Brain Function and Disease (2020B1212060024).

## Additional information

### Funding

| Funder | Grant reference number | Author |
|---|---|---|
| Ministry of Science and Technology of China | the National Key R&D Program of China (2018YFA0108300) | Mingtao Li |
| National Natural Science Foundation of China | U1801681 | Qiaoying Huang Shanshan Ma Mingtao Li |
| National Natural Science Foundation of China | 81771368 | Qiaoying Huang Shanshan Ma Mingtao Li |
| National Natural Science Foundation of China | 31871019 | Qiaoying Huang Shanshan Ma Mingtao Li |
| National Natural Science Foundation of China | 32070959 | Qiaoying Huang Shanshan Ma Mingtao Li |
| Department of Science and Technology of Guangdong Province | the Key Realm R&D Program of Guangdong Province (2018B030337001) | Mingtao Li |
| Department of Science and Technology of Guangdong Province | the Guangdong Provincial Key Laboratory of Brain Function and Disease (2020B1212060024) | Mingtao Li |

The funders had no role in study design, data collection and interpretation, or the decision to submit the work for publication.

### Author contributions

Weizhao Chen, Conceptualization, Data curation, Formal analysis, Methodology, Writing – original draft, Writing – review and editing; Qiongping Zheng, Data curation, Methodology; Qiaoying Huang, Investigation, Validation, Writing – original draft, Writing – review and editing; Shanshan Ma, Funding acquisition, Investigation, Methodology, Project administration, Resources, Supervision, Visualization, Writing – review and editing; Mingtao Li, Conceptualization, Funding acquisition, Project administration, Resources, Supervision, Writing – review and editing

### Author ORCIDs
Weizhao Chen http://orcid.org/0000-0002-6753-4854
Qiongping Zheng http://orcid.org/0000-0002-1989-9234
Shanshan Ma http://orcid.org/0000-0002-3004-9468
Mingtao Li http://orcid.org/0000-0001-5714-9322

### Ethics

All animal experiments were approved and performed in strict accordance with the guidelines by the Institutional Animal Care and Use Committee (IACUC) protocols (No.2018-059) of Sun Yat-Sen University, Guangzhou,China. The protocol was reviewed and approved by the Ethics Committee of Zhongshan School of Medicine(ZSSOM) on Laboratory Animal Care(Permit number: SYSU-IACUC-2018-059). All surgery was performed under isoflurane anesthesia, and every effort was made to minimize suffering.

### Decision letter and Author response
Decision letter https://doi.org/10.7554/eLife.75636.sa1

Author response https://doi.org/10.7554/eLife.75636.sa2

---

## Additional files

### Supplementary files
• Transparent reporting form

### Data availability
All data generated or analysed during this study are included in the manuscript and supporting file; source data files have been provided for all the figures.

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
