## [Editor Report]

In aggregate, we believe that this study provides important new light on the role of PTB1 repression as a potential therapeutic repair strategy. This paper elegantly utilizes a broad array of approaches to demonstrate, in a toxin-based mouse model of Parkinson's disease, that reactive astrocytes fail to convert into neurons such as dopaminergic neurons in brain areas specifically affected in this common neurodegenerative disorder (i.e. the striatum and the substantia nigra) upon repression of the RNA-binding protein PTB1. This finding is of enormous importance since previous studies have reported the reverse, namely that astrocytes can be converted into dopaminergic neurons in response to the repression of PTB1 and have used these results to propose that inhibition of PTB1 by gene therapy could represent new repair strategies for patients with disorders of the nervous system, such as Parkinson’s disease.

---

## [Decision Letter]

**Decision letter after peer review:**

Thank you for submitting your article "Repressing PTBP1 is incapable to convert reactive astrocytes to dopaminergic neurons in a mouse model of Parkinson's disease" for consideration by *eLife*. Your article has been reviewed by 3 peer reviewers, one of whom is a member of our Board of Reviewing Editors, and the evaluation has been overseen by Suzanne Pfeffer as the Senior Editor. The following individual involved in review of your submission has agreed to reveal their identity: Benedikt Berninger (Reviewer #2).

Essential revisions:

All three reviewers expressed a high level of enthusiasm for your work and made important comments/recommendations (see appended reviews) that are, for the most part, straightforward and aimed at strengthening your study. Thus, we would like to see these being addressed before your manuscript can be considered suitable for publication. Moreover, we would like you to also consider the following revisions:

(1) To tone-down your findings throughout including in the title as it was felt that you overstate the implications of this work given the fact that clearly, 6-OHDA does not capture the etiology of PD and that your data leave open the possibility that modulation of ptbp1 could facilitate dopaminergic neurogenesis if tested in other models, by other techniques, over longer time scales, or in greater Ns.

(2) To edit the presentation of the statistics as mentioned by Reviewer #1 throughout.

(3) to revise your discussion. Indeed, we concur with the Reviewers that a more formal discussion would increase the potential impact of your work. In particular, it would be important that the discussion better covers the limitations of the present work and stresses the outstanding questions.

Accordingly, we would recommend that you discuss at least the following two points:

(1) the acute nature of the 6-OHDA model and whether this may elicit cellular alterations, including in astrocytes, distinct from those that may arise in a chronic neurodegenerative process; and

(2) whether AAVs themselves could prevent neuronal conversion as suggested by Johnston et al., (*eLife* 2021). Although we recognize that many studies claiming successful glia-to-neuron conversion have used 6-OHDA and AAVs, we believe that you should discuss the possibility that other models of PD (as already mentioned above) and/or other approaches of knocking down Ptbp1, could potentially yield positive results. For instance, Maimon et al., (Nat Neurosci 2021) used antisense oligonucleotides against Ptbp1 to observe increased neurogenesis in the dentate gyrus, but not so markedly in other brain areas.

*Reviewer #1 (Recommendations for the authors):*

This is a very strong, elegant and straightforward study designed to test the question as to whether compared to quiescent astrocytes, reactive astrocytes may be more amendable to be converted into neurons, and specifically, dopaminergic neurons upon repression of PTB1. As a first step, the authors show that a number of neurons, including dopaminergic neurons, did express GFP in the substantia nigra of mice after injection of AAV-Gfap-EGFP-shPtb1. However, upon combining an elegant lineage tracing method, the authors found no evidence of GFP/NeuN or GFP/TH neurons derived from HA labeled astrocytes in both intact and 6-OHDA lesioned mice. In the latter the authors showed data in support of the reactive state of astrocytes. In light of this data, Chen and collaborators concluded that, in mice, there is no evidence of quiescent or reactive astrocyte conversion into dopaminergic neurons upon repression of PTB1.

This is a very convincing study, extremely well designed and conducted, and shows a set of striking data. The methods are excellent, and the results are well presented and discussed.

*Reviewer #2 (Recommendations for the authors):*

Considering the following points might improve the manuscript

1. It would be useful to show which cells were targeted by the AAV construct at earlier time points, e.g. 7days following infection. There is a general agreement that hGFAP promoter may be leaky and a tighter system would use AAVs with floxed transgenes. However, even if the system lacks in specificity, the authors do show convincingly that Ptbp1 is knocked down in astrocytes. This effect, illustrated in Figure 1B should be quantified.

2. In Figure 1C, the authors should provide 3D visualization of individual neurons to demonstrate co-localization of GFP with NeuN or TH more clearly.

3. The authors should quantitatively compare the number of NeuN and TH positive neurons in w/o 6-OHDA controls to those following scramble or Ptbp1 knockdown treatment in figure 3.

4. Given that the authors suspect that AAV-mediated GFP expression occurs in neurons despite being driven from hGAP promoter, would it make sense to show presence of GFP mRNA in neurons, even if very little, at an time point when GFP fluorescence is still undetectable in neurons but clearly seen in astrocytes? The authors could use RNA Scope towards this. I would expect to see clear signs of low-level expression compared to massive expression in astrocytes.

5. Given that hGFAP driven expression should eventually cease when astrocytes convert, would this have any impact on the interpretation? The authors should analyze whether there are any fate-mapped neurons that seem to lack AAV-mediated GFP expression. Or do the AAV constructs used in Figure 3 lack GFP to allow for TH IF?

6. The left panels in 3B must be controls, but this is not clearly indicated in the figure legend.

7. English should be revised.

8. Regarding the ethical endorsement of animal experiments, more clear information regarding the endorsing body should be provided. Only "university" is mentioned (Care and Use committee of university). Is there any legal number associated to this endorsement?

*Reviewer #3 (Recommendations for the authors):*

Chen et al., highlight a technical confound in previous publications that claim attenuation of PTBP1 results in trans-differentiation of astrocytes into dopamine neurons. While they clearly observe that previous use of recombinant AAV vectors lead to EGFP expression in dopaminergic neurons despite using an astrocyte "specific" promoter, the authors cannot conclude from the presented experiments that attenuation of PTBP1 does not achieve trans-differentiation. This is because the use of AAV could render astrocytes unable to differentiate into dopaminergic neurons in response to successful downregulation of PTBP1. Adding other, AAV independent, means of achieving PTBP1 downregulation in astrocytes in vivo would strengthen this study.

---

## [Author Response]

Essential revisions:All three reviewers expressed a high level of enthusiasm for your work and made important comments/recommendations (see appended reviews) that are, for the most part, straightforward and aimed at strengthening your study. Thus, we would like to see these being addressed before your manuscript can be considered suitable for publication. Moreover, we would like you to also consider the following revisions:(1) to tone-down your findings throughout including in the title as it was felt that you overstate the implications of this work given the fact that clearly, 6-OHDA does not capture the etiology of PD and that your data leave open the possibility that modulation of ptbp1 could facilitate dopaminergic neurogenesis if tested in other models, by other techniques, over longer time scales, or in greater Ns.

We appreciate the editor’s suggestion and have revised our title to ‘Repressing PTBP1 fails to convert reactive astrocytes to dopaminergic neurons in a mouse 6-hydroxydopamine model of Parkinson's disease’.

(2) To edit the presentation of the statistics as mentioned by Reviewer #1 throughout.

The detailed statistical information is provided in our revised manuscript.

(3) to revise your discussion. Indeed, we concur with the Reviewers that a more formal discussion would increase the potential impact of your work. In particular, it would be important that the discussion better covers the limitations of the present work and stresses the outstanding questions.

As suggested by the editor and reviewers, we have added a formal discussion in our revised manuscript that covers the limitations of the present work and the outstanding questions in this field.

Reviewer #2 (Recommendations for the authors):Considering the following points might improve the manuscript1. It would be useful to show which cells were targeted by the AAV construct at earlier time points, e.g. 7days following infection. There is a general agreement that hGFAP promoter may be leaky and a tighter system would use AAVs with floxed transgenes. However, even if the system lacks in specificity, the authors do show convincingly that Ptbp1 is knocked down in astrocytes. This effect, illustrated in Figure 1B should be quantified.

To identify the exact cell type targeted by our AAV construct seven days after AAV infection, we performed immunostaining and GFP^+^ cell number counting in the substantia nigra and striatum, respectively. The results showed that most of the infected cells were astrocytes (AldoC^+^), and a few other cells, such as neurons (NeuN^+^), NG2-glia (NG2^+^), and microglia (Iba-1^+^), were infected (Figure 1—figure supplement 1).

Moreover, as suggested by the reviewer, we have quantified the PTBP1 knockdown efficiency in Figure 1B, and the quantified data are included in the revised manuscript (Figure 1C).

2. In Figure 1C, the authors should provide 3D visualization of individual neurons to demonstrate co-localization of GFP with NeuN or TH more clearly.

To more clearly demonstrate the colocalization of GFP with NeuN or TH, we performed immunofluorescence experiments using anti-GFP, anti-TH, and anti-NeuN antibodies. Images were captured using a LSM780 confocal microscope (Zeiss). The enlarged 3D reconstruction of boxed individual neurons are shown in the lower panel of Figure 1D, F in the revised manuscript.

3. The authors should quantitatively compare the number of NeuN and TH positive neurons in w/o 6-OHDA controls to those following scramble or Ptbp1 knockdown treatment in figure 3.

As suggested by the reviewer, we have quantified the number of NeuN^+^ neurons and TH^+^ dopaminergic neurons in the substantia nigra before and after AAV-sh*Ptbp1* or AAV-shscramble delivery in the 6-OHDA model and found no significant difference between the two groups, indicating that no neurons, including DA neurons, were regenerated in the substantia nigra after astroglial PTBP1 repression (Figure 3D, E).

4. Given that the authors suspect that AAV-mediated GFP expression occurs in neurons despite being driven from hGAP promoter, would it make sense to show presence of GFP mRNA in neurons, even if very little, at an time point when GFP fluorescence is still undetectable in neurons but clearly seen in astrocytes? The authors could use RNA Scope towards this. I would expect to see clear signs of low-level expression compared to massive expression in astrocytes.

We agree with the reviewer that low levels of GFP mRNA may appear in neurons at early time points, and we did observe low GFP protein levels in neurons 1 month after AAV-sh*Ptbp1* delivery, but not AAV-shscramble (Figure 1 D, F). According to a recent study, the coding sequence of some proneural genes, such as *Neurod1* could activate *hGfap* promoter via binding to cis-acting elements of the promoter in neurons (Wang et al., 2021). Therefore, we suppose that the shRNA sequence targeting *Ptbp1* per se might function similar to *Neurod1*, activating the *hGfap* promoter to drive GFP expression in neurons at a relatively low level.

5. Given that hGFAP driven expression should eventually cease when astrocytes convert, would this have any impact on the interpretation? The authors should analyze whether there are any fate-mapped neurons that seem to lack AAV-mediated GFP expression. Or do the AAV constructs used in Figure 3 lack GFP to allow for TH IF?

We understand the reviewer’s concern. Because the *hGfap* promoter activity gradually ceased during AtoN conversion, we chose *Rpl22^HA/HA^* reporter mice to permanently label astrocytes with HA-tag. If AtoN conversion occurrs, astrocyte-derived neurons or dopaminergic neurons should be marked with HA. However, we did not detect any HA^+^NeuN^+^ or HA^+^TH^+^ cells after PTBP1 repression, suggesting that no neurons or DAn were converted from astrocytes.

The two AAV constructs used in this study were attached to EGFP driven by the *hGfap* promoter as illustrated in Figure 1A. We apologize for the misleading comment because GFP images were not provided, which have been included in the revised manuscript (Figure 3C). The fluorescence images in the manuscript were captured on a gray scale and pseudocolored for overlay presentation.

6. The left panels in 3B must be controls, but this is not clearly indicated in the figure legend.

We appreciate the reviewer’s reminder and have revised according to reviewer’s suggestion.

7. English should be revised.

As suggested by the reviewer, we have sent our manuscript to the Editage (www.editage.cn) for English language editing.

8. Regarding the ethical endorsement of animal experiments, more clear information regarding the endorsing body should be provided. Only "university" is mentioned (Care and Use committee of university). Is there any legal number associated to this endorsement?

We supplied detailed ethical endorsement information of animal experiments including permission numbers in the Animal part of the Material and Methods section.

Reviewer #3 (Recommendations for the authors):Chen et al., highlight a technical confound in previous publications that claim attenuation of PTBP1 results in trans-differentiation of astrocytes into dopamine neurons. While they clearly observe that previous use of recombinant AAV vectors lead to EGFP expression in dopaminergic neurons despite using an astrocyte "specific" promoter, the authors cannot conclude from the presented experiments that attenuation of PTBP1 does not achieve trans-differentiation. This is because the use of AAV could render astrocytes unable to differentiate into dopaminergic neurons in response to successful downregulation of PTBP1. Adding other, AAV independent, means of achieving PTBP1 downregulation in astrocytes in vivo would strengthen this study.

We deeply appreciate Reviewer 3’s constructive suggestions. We injected antisense oligonucleotide (ASO) against mouse *Ptbp1* in the substantia nigra of lineage-tracing mice with or without 6-OHDA lesions for 2 months. While astroglial PTBP1 was efficiently repressed after ASO delivery for 2 months (Figure 4—figure supplement 1), no neurons, including DAn expressing astroglial tracing reporter, were detected, indicating that no neurons, including DAn, were converted from quiescent or reactive astrocytes after ASO-*Ptbp1* delivery (Figure 4B, D). Moreover, motor deficits in 6-OHDA lesion mice did not improve 2 months after ASO-*Ptbp1* delivery. These results have been added to the revised manuscript (Figure 4E).

References

Brulet R, Matsuda T, Zhang L, Miranda C, Giacca M, Kaspar BK, Nakashima K, Hsieh J (2017) NEUROD1 Instructs Neuronal Conversion in Non-Reactive Astrocytes. Stem cell reports 8:1506-1515.

Grande A, Sumiyoshi K, Lopez-Juarez A, Howard J, Sakthivel B, Aronow B, Campbell K, Nakafuku M (2013) Environmental impact on direct neuronal reprogramming in vivo in the adult brain. Nature communications 4:2373.

Guo Z, Zhang L, Wu Z, Chen Y, Wang F, Chen G (2014) in vivo direct reprogramming of reactive glial cells into functional neurons after brain injury and in an Alzheimer's disease model. Cell stem cell 14:188-202.

Mattugini N, Bocchi R, Scheuss V, Russo GL, Torper O, Lao CL, Gotz M (2019) Inducing Different Neuronal Subtypes from Astrocytes in the Injured Mouse Cerebral Cortex. Neuron 103:1086-1095 e1085.

Qian H, Kang X, Hu J, Zhang D, Liang Z, Meng F, Zhang X, Xue Y, Maimon R, Dowdy SF, Devaraj NK, Zhou Z, Mobley WC, Cleveland DW, Fu XD (2020) Reversing a model of Parkinson's disease with in situ converted nigral neurons. Nature 582:550-556.

Wan J, Zhao XF, Vojtek A, Goldman D (2014) Retinal injury, growth factors, and cytokines converge on β-catenin and pStat3 signaling to stimulate retina regeneration. Cell Rep 9:285-297.

Wang LL, Serrano C, Zhong X, Ma S, Zou Y, Zhang CL (2021) Revisiting astrocyte to neuron conversion with lineage tracing in vivo. Cell.

Zhou H et al. (2020) Glia-to-Neuron Conversion by CRISPR-CasRx Alleviates Symptoms of Neurological Disease in Mice. Cell 181:590-603.e516.